# Genome Characterisation of the CGMMV Virus Population in Australia—Informing Plant Biosecurity Policy

**DOI:** 10.3390/v15030743

**Published:** 2023-03-14

**Authors:** Joanne Mackie, Paul R. Campbell, Monica A. Kehoe, Lucy T. T. Tran-Nguyen, Brendan C. Rodoni, Fiona E. Constable

**Affiliations:** 1School of Applied Systems Biology, La Trobe University, Melbourne, VIC 3083, Australia; 2Agriculture Victoria Research, Department of Energy, Environment and Climate Action, Melbourne, VIC 3083, Australia; 3Horticulture and Forestry Science, Department of Agriculture and Fisheries, Ecosciences Precinct, Brisbane, QLD 4102, Australia; 4Biosecurity and Sustainability, Department of Primary Industries and Regional Development, South Perth, WA 6151, Australia; 5Plant Health Australia, Canberra, ACT 2600, Australia

**Keywords:** cucumber green mottle mosaic virus, genetic diversity, median-joining network

## Abstract

The detection of cucumber green mottle mosaic (CGMMV) in the Northern Territory (NT), Australia, in 2014 led to the introduction of strict quarantine measures for the importation of cucurbit seeds by the Australian federal government. Further detections in Queensland, Western Australia (WA), New South Wales and South Australia occurred in the period 2015–2020. To explore the diversity of the current Australian CGMMV population, 35 new coding sequence complete genomes for CGMMV isolates from Australian incursions and surveys were prepared for this study. In conjunction with published genomes from the NT and WA, sequence, phylogenetic, and genetic variation and variant analyses were performed, and the data were compared with those for international CGMMV isolates. Based on these analyses, it can be inferred that the Australian CGMMV population resulted from a single virus source via multiple introductions.

## 1. Introduction

*Cucumber green mottle mosaic virus* (CGMMV) is a species of the family *Virgaviridae* and genus *Tobamovirus* [1]. It is a positive-sense, single-stranded, rod-shaped virus with a monopartite genome of 6.4 kb [2]. Four open reading frames (ORFs) encode proteins of 186 K, 129 K, 29 K (movement protein) and 17.3 K (coat protein) [3]. First described in 1935 in England in *Cucumis sativus* (cucumber) [4], CGMMV infects plants in the *Cucurbitaceae* family as well as a number of weeds and wild plants from the *Amaranthaceae*, *Apiaceae*, *Boraginaceae*, *Chenopodiaceae*, *Lamiaceae*, *Portulacaceae* and *Solanaceae* families [5,6,7]. Mechanical and seed transmission are the primary modes of spread [8]. It has also been reported that pollinators such as *Apis mellifera* (European honeybee) can spread the virus while foraging [9].

CGMMV worldwide distribution includes Africa, Asia, Australia, Europe, the Middle East and North America, with the majority of detections having occurred between 1986 and 2016 [8,10]. The first Australian detection of CGMMV occurred in September 2014, when mosaic and mottle symptoms were observed on commercial *Citrullus lanatus* (watermelon) crops near Katherine and Darwin, Northern Territory (NT) [11]. Delimiting surveys reported an additional 26 NT locations with virus-infected cucurbit crops or weeds. By March 2015, eradication was considered technically unfeasible and management practices were instigated. In subsequent years, outbreaks have occurred in Queensland (QLD), Western Australia (WA), South Australia (SA) and New South Wales (NSW), with eradication or management plans being implemented in production areas [12,13,14].

The majority of commercial cucurbit seed used by Australian growers is imported [15]. Prior to 2014, cucurbit propagative material was permitted entry to Australia without any specific disease testing [12]. In 2014, the first report of CGMMV in Australia resulted in the introduction of emergency phytosanitary measures for cucurbit seeds associated with CGMMV (*C. lanatus*, *Cucumis melo*, *Cucumis sativus*, *Cucurbita maxima*, *Cucurbita moschata*, *Cucurbita pepo*, *Lagenaria siceraria*, *Trichosanthes cucumerina* and any hybrid of these species) by the Australian federal government to manage the risk of further CGMMV introductions into Australia [12]. The emergency-measure import conditions for seeds of the listed species stipulate the mandatory testing for CGMMV using an International Seed Testing Association-accredited ELISA protocol on large seed-lot subsamples of 9400 seeds or small seed lots of a 20% subsample. In 2016, Australian laboratories (Elizabeth Macarthur Agricultural Institute (EMAI), New South Wales, and the Crop Health Services (CHS), Agriculture Victoria Research (AVR)) detected CGMMV in 22 out of 631 (3.5%) seed lots, resulting in the exclusion of these seeds from Australia [15]. These seed lots were understood to have originated from Europe, the Middle East, Africa, and North, Central and South America [15], reflecting the extent of CGMMV spread across the globe. Genomic and phylogenetic analysis of the global CGMMV population has shown the distinct clustering of isolates into two major clades [8,16] that align with the geographic regions of Europe and Asia.

The detection of CGMMV in Australia occurred shortly after the detection of CGMMV in California in 2013 [8]. A recent study using sequence and phylogenetic analyses of Californian CGMMV isolates showed that multiple introductions via virus-infected seed had occurred [17]. It is unknown when CGMMV entered Australia and whether there have been multiple introductions. In this study, CGMMV isolates sourced from local detections and post-border seed interceptions were analysed to assess diversity and provide insight into the number of incursions leading to the 2014 Northern Territory outbreak and the subsequent movement of the virus within Australia. Genome analyses were completed using 42 new Australian CGMMV coding sequence complete genomes and 139 CGMMV genomes retrieved from GenBank, and they suggest that a single genotype has been introduced by either single or multiple incursion events.

## 2. Materials and Methods

### 2.1. Virus Isolates

Thirty-nine CGMMV isolates were used to generate genomes for this study (Table 1). Two *C. sativus* isolates collected during outbreaks of CGMMV in Queensland were provided as freeze-dried material by the Department of Agriculture and Fisheries, Brisbane, Queensland, Australia. Twenty hive-collected pollen samples and RNA extracts of *C. lanatus*, *C. lanatus* var. *lanatus*, *Eleusine indica* and *Solanum nigrum* were provided by the Department of Industry, Tourism and Trade (DITT), Northern Territory. Crop Health Services (CHS), Agriculture Victoria Research (AVR), Victoria, provided RNA from four *C. sativus* leaf and fruit samples collected from properties on the Northern Adelaide Plains, South Australia, affected during an outbreak in 2019 and subsequent surveillance in 2020. Four seed interception isolates were sourced from the Australian diagnostic laboratories, EMAI (NSWDPI) and CHS (AVR). Raw sequence data were provided for the original Northern Territory and Queensland detections in *C. lanatus* and for a *C. sativus* isolate collected in the Western Sydney cropping region and a *C. lanatus* isolate from the Sunraysia vegetable growing region, both collected during surveillance activities.

### 2.2. RNA Extraction and RT-PCR Amplification

Total RNA was extracted from each pollen sample using the RNeasy Plant Mini Kit (QIAGEN, Doncaster, VIC, Australia). Starting with 0.05 g of hive-collected pollen, samples were homogenised using approximately 600 µL of 3 mm solid-glass beads (Merck Pty. Ltd., Bayswater, VIC, Australia) and 600 µL QIAGEN RLT buffer. After homogenisation, 300 µL RLT buffer and 10 µL ß-mercaptoethanol were added to the homogenate, and the extraction was completed according to the manufacturer’s instructions. Leaf and fruit samples that had been submitted to Agriculture Victoria’s Crop Health Services diagnostics laboratory and the NSW isolates collected during surveillance were extracted using the RNeasy Plant Mini Kit with a modified lysis buffer [18]. RNA of isolates that were provided by Northern Territory colleagues and of the original NT isolate were extracted using the Isolate II RNA Plant Kit (Bioline (Aust) Pty Ltd., Eveleigh, NSW, Australia) as per the manufacturer’s instructions. The QLD isolate Q6393 was prepared using the Qiagen BioSprint plant DNA Kit (QIAGEN, Doncaster, VIC, Australia) as per the manufacturer’s instructions, omitting the RNase A from the RLT extraction buffer.

Individual samples of imported seed lots, each sample comprising 100 seeds, were crushed and homogenised in a 5 × phosphate-buffered saline containing 0.25% (*v/v*) Tween^®^20 and 2% (*w/v*) polyvinylpyrrolidone 40,000, which was added at a rate of 9 mL of buffer to 1 g of seed. RNA was extracted according to the manufacturer’s instructions using a 100 µL aliquot of homogenate added to 450 µL RLT buffer (RNeasy Plant Mini Kit, QIAGEN, Doncaster, VIC, Australia).

RT-PCR and RT-qPCR tests were conducted using the GoTaq Probe 1-step RT-qPCR System (Promega Corporation, Alexandria, NSW, Australia) according to the manufacturer’s instructions. Extracts were tested for the presence of CGMMV using primers targeting the coat protein [19], the RNase helicase subunit [20] and the movement protein [21] (Table 2). The PCR products were analysed by electrophoresis in 1.5% agarose gel stained with SYBR™ Safe DNA gel stain (Thermo Fisher Scientific, Scoresby, VIC, Australia). Fragment sizes were determined by comparison against the Invitrogen™ 1 Kb plus DNA ladder (Thermo Fisher Scientific, Scoresby, VIC, Australia).

### 2.3. Metagenomic Sequencing and Bioinformatics

Sequencing libraries for 25 isolates were prepared using an Illumina^®^ TruSeq^®^ Stranded Total RNA with Ribo-Zero Plant preparation kit, as described previously [22]. Libraries were sequenced using the Illumina MiSeq with a paired read length of 2 × 250 bp or the Illumina NovaSeq with a paired read length of 2 × 150 bp (see Table 1).

Raw sequence reads generated in this study and those supplied by collaborators were quality-filtered (quality score ≥ 20, minimum read length: 50), adapter sequences were trimmed and read pairs were validated using Trim Galore! [23]. Read pairs were merged using fastp (Version 0.20.0) [24]. De novo assembly was performed with SPAdes (version 3.15.2) [25] using options-rnaviral and -k 127,107,87,67,31 [26]. Assembled contigs of 1000 nt or more were analysed using BLASTn (version 2.9.0) [27]. Trimmed reads were used to map to the CGMMV reference genome (GenBank accession NC_001801.1) and assemble virus contigs of interest using BBMap (version 38.87) [28] with default settings. BCFtools (version 1.12) was used to call consensus sequences from mapping alignments. Final consensus sequences were created and annotated in Geneious Prime (version 2022.2.1) from mapping and contig consensus sequences.

### 2.4. Tiled Amplicon Multiplex PCR and MinION Sequencing and Bioinformatics

Targeted whole genome sequencing (TWG-seq) using a tiled amplicon sequencing and an Oxford Nanopore (ONT) MinION device was used to generate genome sequences for nine CGMMV isolates from pollen, as described previously [22].

### 2.5. Sequence Analysis and Recombination Detection

The CGMMV genome sequences of the 35 Australian isolates from plants, the 4 seed interception isolates and the 9 Australian CGMMV genomes downloaded from GenBank (Table 3) were used to generate a multiple sequence alignment. The 5′- and 3′-untranslated regions (UTRs) were removed, and the full-length coding sequence regions were aligned using MUSCLE (version 3.8.1551) [29] implemented in MEGA-X [30]. Identical sequences were not included in further analysis (see Table 1). Amino acid alignments were produced for the protein coding regions 129 K and 186 K, the movement protein (MP) and the coat protein (CP).

To compare the CGMMV genomes from Australian plants and intercepted seed isolates from this study with genomes of global isolates, all publicly available CGMMV genomes were retrieved from GenBank (Appendix A). Sequences with degenerate or unknown bases were discarded. The untranslated regions were removed from the sequences, and a total of 171 CGMMV isolates, including 132 publicly available sequences from GenBank, were aligned using MUSCLE (version 3.8.1551) [29] implemented in MEGA-X [30]. Sequence Demarcation Tool version 1.2 (SDT version 1.2) [31] was used to generate and visualise the pairwise nucleotide sequence identity matrix. Amino acid percentage similarity matrices were generated using Geneious Prime (version 2022.2.1) and scoring matrix Blosum62 (threshold 0).

RDP4 (Recombination detection program version 4) [32] was used to detect recombination breakpoints in the alignment of all CGMMV full coding sequence genomes. Default settings were used with the seven detection methods: RDP [33], GENECONV [34], Bootscan [35], MaxChi [36], Chimaera [37], SiScan [38], PhylPro [39] and 3 Seq [40]. Recombination signals that were identified by RDP4.9 as potentially arising through evolutionary processes other than recombination were omitted. A Bonferroni-corrected *p*-value < 0.05 for four or more recombination detection methods was considered credible evidence of a recombination event.

### 2.6. Phylogenetic Analysis

Using the best DNA and protein model determined using MEGA-X [30] for each multiple sequence alignment, maximum-likelihood (ML) phylogenetic trees were constructed with 1000 bootstrap replicates, and the resulting trees were visualised in Interactive Tree of Life (iTOL) version 6.5.8 [41].

Bayesian inference was performed using BEAST version 1.10.4 [42]. Four replicated runs of 50 million Monte Carlo Markov chains (MCMCs) were carried out using the TN93 substitution model, the gamma and invariant sites heterogeneity model, tip dates (sampling dates), and a strict molecular clock model on the coding region nucleotide sequence alignment. The runs were assessed for convergence with Tracer (version 1.7.2) [43], and the first 25% of each run was discarded as burn-in. The resulting trees were merged using LogCombiner and summarised using TreeAnnotator, which are both part of the BEAST package [44]. The maximum-clade-credibility (mcc) tree was visualised using iTOL (version 6.5.8) [41].

Median-joining network analysis of variants was carried out using PopART 1.7 [45,46] for the coat protein and movement protein nucleotide sequences of all CGMMV isolates. Full-length coding sequences were analysed for clades containing Australian and seed interception isolates.

### 2.7. Genetic Variation

Analysis of genomic variation was carried out on the 171 CGMMV isolate data set for all coding sequence regions. DnaSP Ver. 6.12.03 [47] sliding window analyses were used to evaluate the number of polymorphic sites, the total number of mutations, the nucleotide diversity index Pi and the number of variants. Neutrality tests, Tajima’s D indices, Fu and Li’s D*-test statistics, and Fu and Li’s F*-test statistics were also calculated. Statistical significance for each was evaluated under a null hypothesis.

## 3. Results

### 3.1. RT-PCR and RT-qPCR

Conventional and real-time RT-PCR was performed on 37 RNA extracts from the Australian plant and seed interception isolates, noting that five isolates were provided as raw sequencing data (Table 1). CGMMV was detected from the RNA extracts using at least one PCR test: 32/37 extracts tested positive in the coat protein RT-PCR assay [19], 34/37 in the RNase helicase subunit RT-PCR assay [20] and 35/36 in the movement protein RT-qPCR assay [21] (Appendix A). NT isolate 24,501 (NT-21014-03) was originally tested by collaborators, using the two conventional RT-PCR assays. The RNA provided was used to prepare the sequencing library.

### 3.2. Genome Sequences

The raw sequence reads obtained for the 42 samples that were sequenced in this study using MiSeq, NovaSeq, or MinION (Table 1) ranged from 4698 to 20,818,214 per sample (Table 1 and Appendix A). After trimming, 4371–20,779,456 reads remained for de novo assembly and reference mapping. De novo assembly produced 1–138 virus contigs for the 33 CGMMV isolates sequenced using MiSeq and NovaSeq. Mapped reads for all sequencing methods ranged from 217 to 16,914,482 (Appendix A). Consensus sequences were generated for 42 isolates from Australia (*n* = 38) and seed interceptions (*n* = 4) with lengths of 6342–6424 nt. Coding sequence regions were obtained for all isolates and used for further analysis. The consensus sequences generated in this study for 39 isolates have been deposited in GenBank (accession numbers OQ198372–OQ198410).

### 3.3. Sequence Analysis and Recombination Analysis

SDT analysis (Figure 1) of the 35 new Australian CGMMV isolates, 4 seed interception CGMMV isolates and the 9 publicly available Australian CGMMV genomes showed that the full coding regions of all Australian isolates shared a high level of nucleotide pairwise identity, ranging from 99.6% to 99.9%. The seed interception isolates SI_2016_01 and SI_2016_02 also shared 99.6 to 99.9% nucleotide identity with Australian isolates. Seed isolates SI_2015_01 and SI_2018_01 shared 98.9–99.2% and 98.4–98.9% pairwise identity, respectively, with the Australian and other seed samples. Amino acid alignments of each of the four coding regions showed minimal differences, with 129 K, 186 K, MP and CP sequences sharing 99.9–100%, 99.8–100%, 100% and 99.4–100% amino acid similarity, respectively (Appendix A).

Comparison with global isolates using the full coding region alignment showed that Australian isolates shared over 99% nucleotide identity with isolates from Canada (GenBank accession KP772568.1), Israel (GenBank accessions KF155229.1 and KF155230.1), the Netherlands (GenBank accessions MH271419.1, MH271420.1 and MH271421.1) and the USA (GenBank accession MH271442.1) (Figure 2, Appendix A).

The recombination analysis of all the CGMMV genomes using RDP4 did not detect any recombinants.

### 3.4. Phylogenetic Analysis

Bayesian maximum-clade-credibility trees were constructed using the full coding region nucleotide sequences of the Australian, seed interception and global isolates sourced from GenBank. The mcc tree inferred for the 171 CGMMV isolates (Figure 3) produced two major clades with greater than 95% posterior probability support and is consistent with previous studies [8,16]. Subclades were defined in this study and were based on posterior probability values as follows. Clade 1 consists of two sub-clades with isolates from France, Latvia, the Netherlands, Russia and Spain in clade 1A and from Germany, Israel, Japan, Kuwait, the Netherlands and the USA in clade 1B. Clade 2 contains four subclades that have been designated 2A, 2B, 2C and 2D (Figure 3). Sub-clade 2A contains single isolates from Israel (KF155231.1) and Thailand (MH271423.1). Sub-clade 2B contains seven isolates from Canada and single isolates from China and Japan. Sub-clade 2C contains isolates originating from Australia, Bulgaria Canada, France, Greece, India, Israel, the Netherlands and the USA (Figure 3). Sub-clade 2D isolates all originated from Asia (China, Japan, Korea, South Korea, Taiwan and Thailand). All Australian isolates and three seed interception isolates are members of sub-clade 2C. The seed interception isolate SI_2018_01 aligns with Asian isolates in sub-clade 2D. Posterior probability support for clades and sub-clades is high (>95). Support for further clades in sub-clade 2C is low (<50).

The mcc tree generated for the Australian and seed interception isolates (Figure 4) shows well-supported clades consisting of all Australian isolates and two seed interception isolates, SI-2016-01 and SI-2016-02, clustering separately to seed isolates SI-2015-01 and SI-2018-01. The Australian clade separates into two clusters; however, posterior probability support for these clusters is low (<60). Cluster I contains all SA and WA genomes, together with two genomes from each of NT and QLD and one genome from NSW. Within Cluster 2, two seed interception genomes and the remaining Australian genomes group together. Seed isolates SI_2015_01 and SI_2018_01 form a separate cluster.

The full coding sequences of the CGMMV isolates clustering in sub-clades 2B and 2C of the mcc tree (Figure 3) were used to generate a median-joining network (Figure 5). All Australian isolates and seed interception isolates (GenBank accessions OQ198399, OQ198400 and OQ198410) are located in Clusters 1, 2 and 3. Cluster 1 includes isolates MH427279.1 (CGMMV-NT), NSW-2019-01, NSW-2020-01, NT-2014-02, NT-2014-03, NT-2015-01, NT-2016-01, NT-2016-02, NT-2017-01, NT-2017-02, NT-2017-03, NT-2019-01, NT-2019-02, NT-2019-03, NT-2019-04, NT-2019-05, NT-2019-06, NT-2019-07, NT-2019-08, NT-2019-09, NT-2019-10, NT-2020-01, NT-2020-02, NT-2020-03, NT-2020-04, NT-2020-05, QLD-2018-01, seed interception isolate SI-2015-01 and European isolate CG015 (MH271421.1). Cluster 2 contains isolates KY115174.1 (WA-1), MW430119.1 (WA-2), MW430120.1 (WA-3), MW430121.1 (WA-4), MW430122.1 (WA-5), MW430123.1 (WA-6), MW430124.1 (WA-7), MW430125.1 (WA-8), QLD_2019_01, QLD_2019_02, SA_2019_01, SA_2019_02, SA_2019_03, SA_2019_04 and SA_2020_01. Cluster 3 contains European isolate CG013 (MH271419.1); Middle Eastern isolate CG006 (MH271412.1); North American isolates ABCA13-01 (KP772568.1), CG002 (MH271408.1), CG004 (MH271410.1), CG036 (MH271441.1) and CG045 (MT184941.1); Australian isolates NT_2015_02 and QLD_2015_01; and seed interceptions SI_2016_01 and SI_2016_02. Cluster 4 contains North American isolates located in sub-clade 2B, and Clusters 5 and 6 contain North American isolates located in sub-clade 2C.

Variant analysis and median-joining networks generated for all coat protein and movement protein sequences from 171 CGMMV isolates (Figure 6) revealed 65 and 84 variants, respectively. Seven of the 65 coat protein variants include Australian isolates. Multilocation variants 29 and 41 include 10 Australian, 1 European and 13 North American isolates and 20 Australian, 2 European and 2 American isolates and 2 isolates of unknown origin, respectively. The remaining five variants, 58, 60–62 and 64, are exclusively Australian isolates. Thirteen of the 84 movement protein haplotypes include Australian isolates. Variant 42 includes five Australian, one European and ten North American isolates; variant 46 contains four Australian; one each from Europe, the Middle East and North America; and two of unknown origin.

### 3.5. Genomic Variation

DnaSP analysis was carried out on 171 CGMMV isolates for the full coding sequence region and the 129 kDa, 186 kDa, MP and CP regions (Table 4). Sequences were grouped and analysed by collection location. For the 44 Australian sequences, 137 of 6188 sites were found to be polymorphic, and the average number of nucleotide differences, k, was calculated to be 19.14905. Nucleotide diversity was low across all regions analysed. Negative Tajima’s D values, indicative of low frequency variation within a population and implying population growth or positive selection, were observed for Asian, Australian and North American isolates across all coding sequence regions, and these were always statistically significant for the Asian isolates. Fu and Li’s D- and F-test values, which also indicate positive selection and population growth, were always negative and statistically significant for Asian, Australian and North American isolates.

## 4. Discussion

The detection and subsequent spread of CGMMV in Australia led to the implementation of emergency measures in 2014 to prevent the risk of further introductions via imported seeds associated with CGMMV. Prior to this, cucurbit seed was not screened for the virus, and questions have arisen regarding the number of introductions that could have occurred pre-2014. Based on the analysis presented here, the introduction of CGMMV into Australia is related to the reporting of CGMMV in North America [17,48,49] and new host reports in Israel [50] and Bulgaria [51], which occurred during the same period (2013–2015) [11]. These isolates are all located in sub-clade 2C (Figure 3) and either share a haplo-group or are within 5 to 10 single nucleotide polymorphisms across the CGMMV genome of Australian variants. The high nucleotide identity between these isolates suggests a shared origin and signifies a global spreading event that is likely to have been facilitated by international seed movement.

Examination of the Australian population of CGMMV using sequence, phylogenetic and genetic variation analyses offers a few possible scenarios for introduction into the country. The first scenario is that there has been a single introduction and then spread via seed, seedling or other mechanical means to several locations and hosts in Australia. The second scenario is that there has been a minimum of two introductions of CGMMV into separate locations in Australia, which may have a common origin offshore.

Support for the first scenario is present in the sequence, phylogenetic and genetic variation analyses of all Australian CGMMV isolates. Full coding sequence genomes are highly similar, with less than 0.5% difference in pairwise nucleotide identity and at most a 0.62% difference in amino acid similarity. Australian variants in the median-joining networks for the coat and movement proteins are separated by 1–2 single nucleotide variants at most. Examination of genetic diversity shows a high variant diversity (Vd) and low nucleotide diversity and reflects the abundance of singletons in the Australian genomes. The negative Tajima’s D and Fu–Li values are statistically different from those expected from a neutral evolution model and are indicative of population expansion or positive selection [52,53,54]. The tip-dated mcc tree estimated using global and Australian CGMMV isolates shows strong posterior probability support for sub-clade 2C containing all Australian isolates, but this support is reduced (<50) at the next split in the tree (Figure 3). Together, these results relating to the diversity of CGMMV in Australia suggest a rapid and recent population expansion that could have occurred following a single introduction.

The second proposed scenario considers the possibility of multiple introductions from the same or similar sources. Whilst sequence analysis of the Australian population does reveal a high level of nucleotide and amino acid similarity, examination of the full coding sequence median-joining network for sub-clades 2C and 2D (Figure 5) shows a partitioning of Australian isolates. This division is also shown in the Australian and seed interception mcc tree (Figure 4), with Clusters 1 and 2 each representing a separate event; however, posterior probability support for this is low. The variant analysis for both the CP and MP indicates the presence of two, well-populated haplo-groups. CP variant 29 and MP variant 42 are both made up of sequences from Australia (*n* = 10 and *n* = 5), North America (*n* = 13 and *n* = 10) and Europe (one each), with common sequences across both haplo-groups. Other sequences from Australia, Europe, the Middle East and North America are all CP variant 41 and MP variant 46. However, the degree of separation between variants does not necessarily support different origins for each introduction. Within Cluster 1, it could also be suggested that a third introduction event may have occurred, represented by the two NT isolates in the upper clade of the tree (Figure 4). However, since these are two of the earliest isolates that were found, it could be that they represent an early variant from which the other Cluster 1 variants emerged.

It is possible that the larger clade within Cluster 1, containing isolates from NSW, QLD, SA and WA (Figure 4), emerged from a similar source of seed, rootstocks or nursery-produced plants. The distribution of virus-contaminated seeds or cucurbit seedlings on infected rootstocks could explain the movement of CGMMV to diverse locations and growing conditions, as reported in an analysis of CGMMV in WA [55], and account for the range of hosts represented in Group 1. Rootstocks, such as *C. moschata* or the hybrid *C.moschata × C. maxima*, are commonly used for commercial crops of watermelons, melons and cucumbers [56]. Quality cucurbit rootstocks can enhance scion growth and provide resistance to soilborne diseases, such as *Fusarium* wilt, *Verticillium* wilt and gummy stem blight [56,57], and while there are benefits to production, there is increased risk of virus introduction via seeds, seedlings and the grafting process.

Alternatively, the stable and infectious nature of CGMMV virions offer a range of transmission pathways within propagation and production settings [8,58]. Establishment of CGMMV after introduction can lead to further spread by farm machinery, grafting tools, seed trays, clothes, shoes and hands [8,59,60]. Dispersal of CGMMV can continue with subsequent planting in soils contaminated with virus-positive plant debris [59,61] and irrigation systems, such as drip and flow irrigation [59]. Movement of virus-positive fresh produce and associated contaminated surfaces can introduce inoculum into post-harvest settings and, in turn, back into production areas. The spread of CGMMV to SA has been linked to infected properties in Geraldton, WA; however, the transmission pathway is unknown [13].

The Queensland isolate QLD-2015-01 (OQ198392) located in the upper clade of Cluster 2 (Figure 4) was detected as a result of seed trace-back following the NT outbreak in 2014 [62]. In this case, seedlings were generated in a QLD propagation facility using seed from the NT and subsequently cultivated on the Charters Towers property. The NSW cucumber isolate NSW-2019-01 (OQ198372) was collected from a glasshouse facility in Western Sydney [63]; however, the source of this outbreak is not clear-cut, with contaminated seeds, seedlings, shared cultivation equipment, soil or water all potential sources of inoculum.

The remaining Cluster 2 isolates resulted from pollinator movement of virus and infection of weeds either by pollen or contact with virus-infected crop debris. The detection of CGMMV in *Apis mellifera* collected from commercial beehives in June 2014 [64] indicates that the virus was present in the NT for a period prior to being detected on crops in September 2014. CGMMV genome sequences from pollen sampled from hives between 2017 and 2020 do not deviate significantly from the first introductions, and accumulated mutations may have arisen from adaptation to an increased host range and changes to environmental conditions [65]. It is worth noting that all pollen samples were collected from different hives across the four-year surveillance period, and it is unclear whether the sequenced virus has been hive-stored for a long period or recently collected by bees. If the latter is the case, then this could be indicative of the presence of the virus population in the bee foraging environment.

Previous studies have demonstrated the involvement of pollinators in the movement of CGMMV during foraging [9,66]. Furthermore, viable CGMMV was detected in adult bees, pollen and honey sampled from apiaries during the 2014 incursion response [67], and subsequent work investigating the role of honey bees in CGMMV epidemiology showed that honey bees can introduce the virus into healthy flowers, resulting in disease symptoms [68]. Viruses from infected plants or from positive beehives both produced symptoms in *C. lanatus* [68]. The range of non-crop species present in Cluster 2 also points to the role that pollinators and green bridges may play in the recurrence and spread of CGMMV infection in a region, highlighting the importance of weed control for disease management for growers. A range of weed species in the *Cucurbitaceae*, *Euphorbiaceae*, *Lamiaceae* and *Solanaceae* families are susceptible to CGMMV [7], and associated seeds have the potential to be a source of virus infection for future crops; however, further studies are needed to examine virus viability and infection rates across different weed hosts.

The analysis of genetic diversity, sequence similarity and phylogenetic relatedness presented in this study supports the presupposition of a common offshore source of CGMMV present in Australia. Support for the multiple-introduction scenario is diminished by the phylogenetic analysis and the lack of support for multiple clades within the Australian population. Examination of the movement of seeds, seedlings, and pre- and post-harvest equipment and produce within Australia provides sufficient evidence for the modes of spread of CGMMV into cucurbit growing regions.

The global spread of CGMMV has been accelerated by the international movement of virus-infected seeds [8], and the outbreaks in Australia and California in 2013–2016 are associated with imported cucurbit seeds [15,17]. Unlike California, where the diversity of the CGMMV population was shown to be the result of multiple introductions [17], the low level of diversity in this study indicates that there was one introduction of CGMMV into Australia, which likely occurred around the time of a global spreading event. It is unlikely there have been further introductions since the introduction of emergency phytosanitary measures for cucurbit seeds associated with CGMMV in 2014, highlighting their effectiveness in mitigating risk to the Australian vegetable and melon industries. Between 2017 and 2022, CHS (AVR) detected CGMMV in 0.5–2.4% of seed lots tested, indicating that contaminated seed continues to circulate on the global market. The current biosecurity import controls on cucurbit seed continue to be a necessary and effective measure against the introduction of new variants, and the continuation of these regulations is vital to the Australian vegetable industries. Resourcing can focus on production and post-harvest systems to manage farm hygiene, cultivation and pollination practices, and weed control, while risk-management measures are in place at the Australian border. This genomic dataset generated for the current Australian CGMMV population provides a baseline for comparison with future detections in Australian vegetable-growing regions.

## Figures and Tables

**Figure 1 viruses-15-00743-f001:**
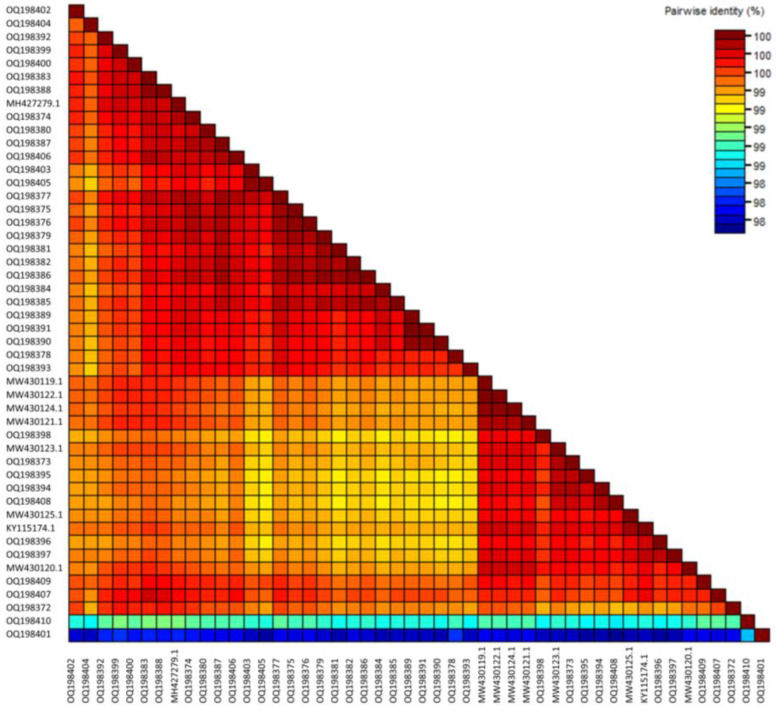
Sequence Demarcation Tool matrix showing percent pairwise nucleotide similarity for the full coding sequence regions of cucumber green mottle mosaic virus genomes from Australian and seed interception samples.

**Figure 2 viruses-15-00743-f002:**
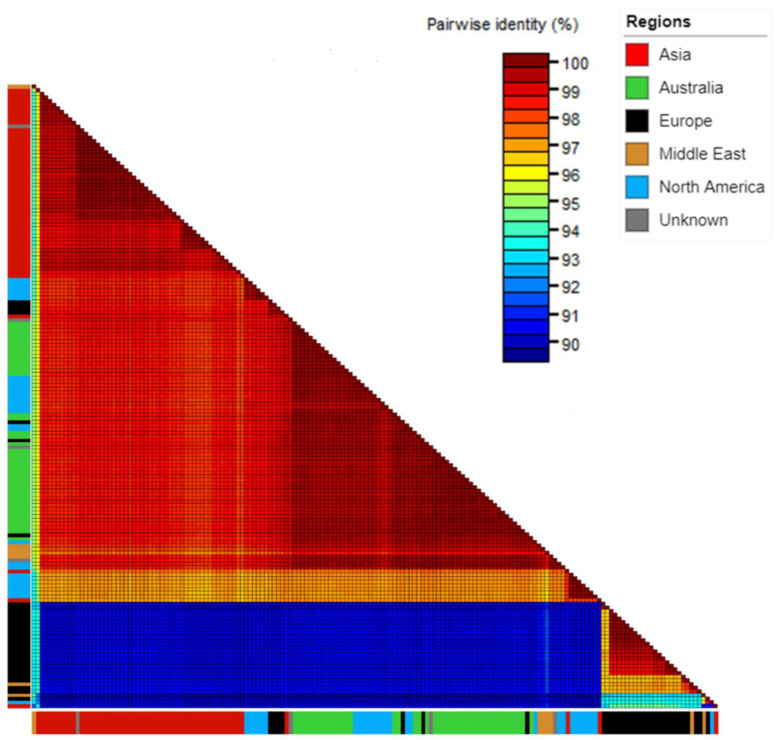
Sequence Demarcation Tool matrix showing percent pairwise nucleotide similarity for the full coding sequence region genomes of Australian, seed interception and global cucumber green mottle mosaic virus isolates. Colour strips on edges denote isolate collection region: Asia—red, Australia—green, Europe—black, Middle East—orange, North America—blue, and unknown isolates (seed interceptions)—grey.

**Figure 3 viruses-15-00743-f003:**
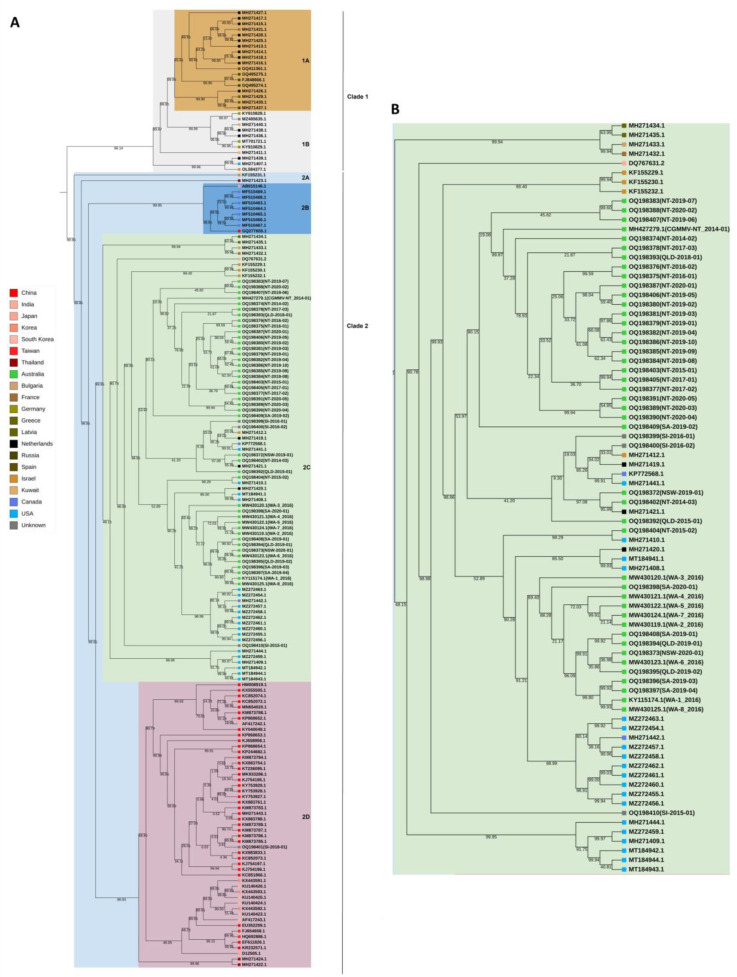
(**A**) Bayesian maximum-clade-credibility (mcc) tree estimated for Australian seed intercepts and publicly available cucumber green mottle mosaic virus (CGMMV) genomes using the complete coding region nucleotide sequences. Background colours identify the main clades. Key identifies country of origin for each accession. (**B**) Clade 2C containing all Australian isolates is presented. The mcc was created using BEAST v1.10.4 [44] using analysis settings: model—Tamura–Nei; rates among sites—gamma distributed with invariant sites. Tip times correspond to the virus sampling date. The posterior probability values are shown on tree branches.

**Figure 4 viruses-15-00743-f004:**
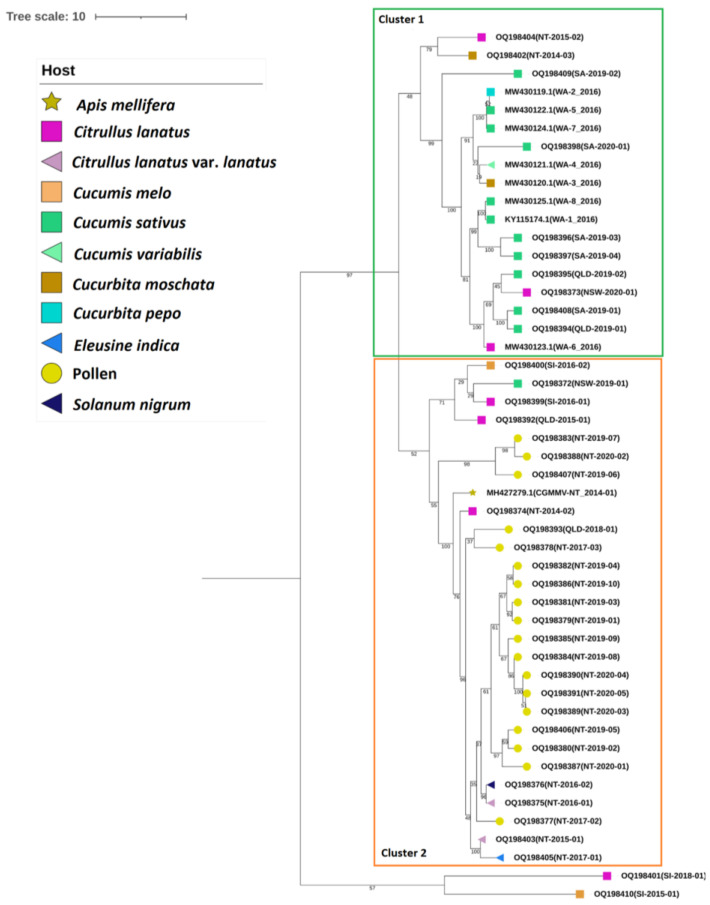
Bayesian maximum-clade-credibility tree estimated for the complete coding region nucleotide sequences of all Australian cucumber green mottle mosaic virus (CGMMV) isolates in sub-clade 2C and seed interception isolates in sub-clades 2C and 2D. The tree was created using BEAST v1.10.4 [44] using the following analysis settings: model—Tamura–Nei; rates among sites—gamma distributed with invariant sites. Tree scale—branch lengths measured by number of substitutions per site. Branch values are posterior probabilities of having a changepoint. Isolate hosts are shown using node symbols (key included).

**Figure 5 viruses-15-00743-f005:**
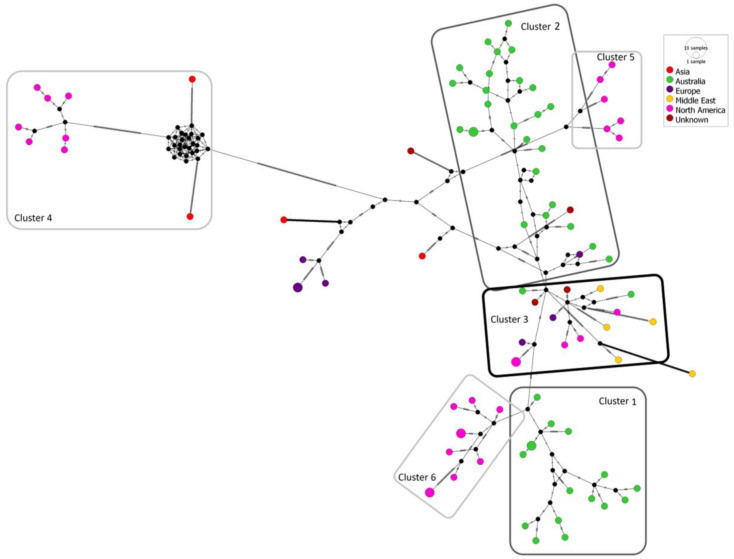
Median-joining network constructed with PopART 1.7, using the full coding sequence of the 80 cucumber green mottle mosaic virus isolates from Australia and other global regions that are placed in Groups III, IV and V of the Bayesian maximum-clade-credibility tree (Figure 4). Node size is proportional to the number of variants. Sequence node colours correspond to the collection regions of isolates. Hatch marks represent the number of single nucleotide variants between black nodes. Cluster 1 accessions: MH271421.1 (CG015), MH427279.1 (CGMMV-NT), NSW-2019-01, NSW-2020-01, NT-2014-02, NT-2014-03, NT-2015-01, NT-2016-01, NT-2016-02, NT-2017-01, NT-2017-02, NT-2017-03, NT-2019-01, NT-2019-02, NT-2019-03, NT-2019-04, NT-2019-05, NT-2019-06, NT-2019-07, NT-2019-08, NT-2019-09, NT-2019-10, NT-2020-01, NT-2020-02, NT-2020-03, NT-2020-04, NT-2020-05, QLD-2018-01 and SI-2015-01. Cluster 2 accessions: KY115174.1 (WA-1), MW430119.1 (WA-2), MW430120.1 (WA-3), MW430121.1 (WA-4), MW430122.1 (WA-5), MW430123.1 (WA-6), MW430124.1 (WA-7), MW430125.1 (WA-8), QLD_2019_01, QLD_2019_02, SA_2019_01, SA_2019_02, SA_2019_03, SA_2019_04 and SA_2020_01. Cluster 3 accessions: KP772568.1, MH271408.1, MH271410.1, MH271412.1, MH271419.1, MH271441.1, MT184941.1, NT_2015_02, QLD_2015_01, SI_2016_01 and SI_2016_02. Cluster 4 accessions: AB015146.1, GQ277655.1, MF510463.1, MF510464.1, MF510465.1, MF510466.1, MF510467.1, MF510468.1, MF510469.1 and MZ272459.1. Cluster 5 accessions: MH271409.1, MH271444.1, MT184942.1, MT184943.1 and MT184944.1. Cluster 6 accessions: MH271442.1, MZ272454.1, MZ272455.1, MZ272456.1, MZ272457.1, MZ272458.1, MZ272460.1, MZ272461.1, MZ272462.1 and MZ272463.1.

**Figure 6 viruses-15-00743-f006:**
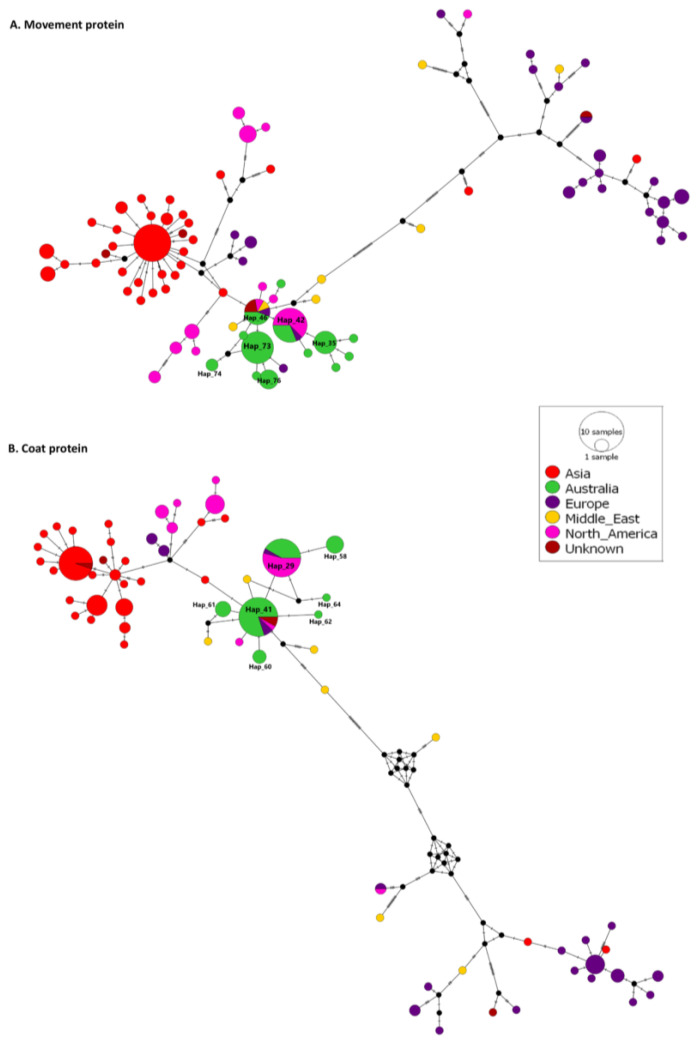
Median-joining networks constructed with PopART 1.7 for movement (**A**) and coat (**B**) proteins using global and Australian and seed interception cucumber green mottle mosaic virus isolates. Node size is proportional to the number of variants. Sequence node colours correspond to the collection regions of isolates. Hatch marks represent the number of single nucleotide variants between nodes.

**Table 1 viruses-15-00743-t001:** The Australian and seed interception cucumber green mottle mosaic virus (CGMMV) isolates used to generate genome sequences for this study.

Sample Name	Isolate Label	Collection Location	Collection Year	Host	Data Source
AWM0504	NSW-2019-01	New South Wales	2019	*Cucumis sativus*	R
NSW3-35	NSW-2020-01	New South Wales	2020	*Citrullus lanatus*	R
VPRI43306	NT-2014-02	Northern Territory	2014	*Citrullus lanatus*	R
24501	NT-2014-03	Northern Territory	2014	*Cucurbita moschata*	M
25811	NT-2015-01	Northern Territory	2015	*Citrullus lanatus* var. *lanatus*	M
26597	NT-2015-02	Northern Territory	2015	*Citrullus lanatus*	M
30467	NT-2016-01	Northern Territory	2016	*Citrullus lanatus* var. *lanatus*	X1
30468	NT-2016-02	Northern Territory	2016	*Solanum nigrum*	X1
32031	NT-2017-01	Northern Territory	2017	*Eleusine indica*	M
M2-3A	NT-2017-02	Northern Territory	2017	Pollen	X1
M3-3A	NT-2017-03	Northern Territory	2017	Pollen	X1
CCP1-A	NT-2019-01	Northern Territory	2019	Pollen	X2
CCP1	NT-2019-02	Northern Territory	2019	Pollen	X2
CCP3	NT-2019-03	Northern Territory	2019	Pollen	X1
CCP3-A	NT-2019-04	Northern Territory	2019	Pollen	X1
CCP-A	NT-2019-05	Northern Territory	2019	Pollen	M
DVP-A	NT-2019-06	Northern Territory	2019	Pollen	M
DWP1	NT-2019-07	Northern Territory	2019	Pollen	X1
DWP2	NT-2019-08	Northern Territory	2019	Pollen	X1
DWP-A	NT-2019-09	Northern Territory	2019	Pollen	X2
RDP1	NT-2019-10	Northern Territory	2019	Pollen	X1
DW20P1	NT-2020-01	Northern Territory	2020	Pollen	X1
DW20P3	NT-2020-02	Northern Territory	2020	Pollen	X1
RH20P2	NT-2020-03	Northern Territory	2020	Pollen	X1
RH20P3	NT-2020-04	Northern Territory	2020	Pollen	X1
RHRSP3	NT-2020-05	Northern Territory	2020	Pollen	X1
Q6393	QLD-2015-01	Queensland	2015	*Citrullus lanatus*	R
QDD2P-1	QLD-2018-01	Queensland	2018	Pollen	X1
J5276	QLD-2019-01	Queensland	2019	*Cucumis sativus*	X1
P2B2-1	QLD-2019-02	Queensland	2019	*Cucumis sativus*	X1
19-03205-1	SA-2019-01	South Australia	2019	*Cucumis sativus*	M
19-03206-1	SA-2019-02	South Australia	2019	*Cucumis sativus*	M
19-03771-Fruit	SA-2019-03	South Australia	2019	*Cucumis sativus*	X1
19-03771-Leaf	SA-2019-04	South Australia	2019	*Cucumis sativus*	X1
SA-20-02243	SA-2020-01	South Australia	2020	*Cucumis sativus*	R
N2_2015	SI-2015-01	Seed interception	2015	*Cucumis melo*	M
N1_2016	SI-2016-01	Seed interception	2016	*Citrullus lanatus*	X1
N3_2016	SI-2016-02	Seed interception	2016	*Cucumis melo*	X1
18-04251-76	SI-2018-01	Seed interception	2018	*Citrullus lanatus*	X1

The “Sample name” is the provider identifier and the “Isolate label” is used for sequence and phylogenetic analysis. Collection location includes seed isolates intercepted at the Australian border. Pollen isolates were sourced from hive-collected pollen collected from mixed plant species. Data source denotes whether the isolate was sequenced using the Illumina NovaSeq (X1) or MiSeq (X2) platforms, CGMMV tiled-amplicon multiplex PCR and ONT MinION sequencing (M), or whether raw HTS data were provided (R).

**Table 2 viruses-15-00743-t002:** Primers used for RT-PCR and RT-qPCR detection of cucumber green mottle mosaic virus (CGMMV).

Primer	Sequence 5′–3′	Size (bp)	Target	Reference
CGMMV-CPF	GATGGCTTACAATCCGATCAC	496	Coat protein	[19]
CGMMV-CPR	CCCTCGAAACTAAGCTTTCG			
CGMMV-RHSF	ATGGCAAACATTAATGAACAAAT	1100	RNase helicase subunit	[20]
CGMMV-RHSR	AACCACACAGAAAACGTGGC			
CGMMV-RZF	GTGGTTTCTGGTGTATGGAACGTA		Movement protein	[21]
CGMMV-RZR	GGTGGCGGGAGCTGAAAA			
CGMMV-RZP	[FAM]-CACCCCTACAGGATTC–[NFQMGB]			

**Table 3 viruses-15-00743-t003:** Australian cucumber green mottle mosaic virus (CGMMV) genomes obtained from GenBank and used for comparison with new Australian genomes.

Accession	Isolate	Collection Location	Collection Date	Host
KY115174.1	WA-1	Geraldton, Western Australia	2016	*Cucumis sativus (*Cucumber)
MW430119.1	WA-2	Geraldton, Western Australia	2016	*Cucurbita pepo (*Zucchini)
MW430120.1	WA-3	Carnarvon, Western Australia	2016	*Cucurbita moschata* (Butternut pumpkin cv. Jacqueline)
MW430121.1	WA-4	Carnarvon, Western Australia	2016	*Cucumis variabilis*
MW430122.1	WA-5	Perth, Western Australia	2016	*Cucumis sativus* (Cucumber cv. Ritoral)
MW430123.1	WA-6	Kununurra, Western Australia	2016	*Citrullus lanatus* (Watermelon)
MW430124.1	WA-7	Carnarvon, Western Australia	2016	*Cucumis sativus* (Slicer cucumber)
MW430125.1	WA-8	Geraldton, Western Australia	2016	*Cucumis sativus* (Cucumber cv. Ritoral)
MH427279.1	CGMMV-NT	Northern Territory	2014	*Apis mellifera* (European honeybee)

**Table 4 viruses-15-00743-t004:** Summary of genetic variation, polymorphism, and neutrality test statistic values for cucumber green mottle mosaic virus (CGMMV) full coding region, 129 kDa, 186 kDa, MP and CP sequences for 171 CGMMV isolates by collection location.

Location	Region	*n*	Sites	S	Eta	Var	Vd	Pi	k	Tajima’s D	FuLiD *	FuLiF *	FuFs
Asia	Full	55	6188	1046	1106	55	1.000	0.014	84.51	−2.353 **	−1.617 **	−1.543 **	−14.681
Australia	Full	44	6188	137	138	42	0.998	0.003	19.15	−1.444	−1.682 *	−1.732 *	−23.635
Europe	Full	31	6188	1095	1161	29	0.996	0.049	304.60	0.187	−1.407	−1.484	1.589
Middle East	Full	7	6188	862	895	7	1.000	0.060	371.00	0.092	−0.504	−0.513	2.849
North America	Full	29	6188	817	836	26	0.993	0.022	138.84	−1.366	−1.332 *	−1.402 *	1.085
Unknown	Full	5	6188	687	691	5	1.000	0.046	287.70	−1.014	−0.367	−0.367	3.347
Asia	129 k	55	3435	615	643	51	0.995	0.015	52.49	−2.261 **	−1.370 *	−1.274 **	−11.487
Australia	129 k	44	3435	84	84	39	0.994	0.003	10.47	−1.645#	−1.462 *	−1.483 *	−27.518
Europe	129 k	31	3435	660	701	29	0.996	0.055	187.39	0.264	0.182	−0.022	0.442
Middle East	129 k	7	3435	522	545	7	1.000	0.065	224.14	0.045	−1.207	−1.217	2.331
North America	129 k	29	3435	511	522	25	0.990	0.025	86.84	−1.359	0.2107 *	0.046 *	0.67
Unknown	129 k	5	3435	420	423	5	1.000	0.051	176.00	−1.018	−0.720	−0.720	2.848
Asia	186 k	55	4947	910	960	54	0.999	0.015	73.00	−2.358 **	−1.635 *	−1.515 **	−13.54
Australia	186 k	44	4947	117	117	41	0.996	0.003	15.90	−1.484	−1.737 *	−1.751 *	−24.354
Europe	186 k	31	4947	916	972	29	0.996	0.052	258.99	0.251	0.197	−0.014	1.207
Middle East	186 k	7	4947	727	756	7	1.000	0.063	311.81	0.062	−1.172	−1.184	2.671
North America	186 k	29	4947	704	720	25	0.990	0.024	120.25	−1.351	0.226 *	0.054 *	1.669
Unknown	186 k	5	4947	585	588	5	1.000	0.050	245.40	−0.998	−0.717	−0.717	3.186
Asia	MP	55	795	84	88	31	0.879	0.008	6.20	−2.37 **	−1.1685 *	−1.396 *	−15.057
Australia	MP	44	795	13	14	13	0.853	0.003	2.13	−1.047	−1.098#	−1.363 #	−4.773
Europe	MP	31	795	110	114	24	0.983	0.034	27.27	−0.170	−0.721	−0.824	−1.719
Middle East	MP	7	795	79	80	7	1.000	0.043	34.14	0.266	−1.628	−1.628	0.242
North America	MP	29	795	69	71	12	0.860	0.015	12.07	−1.265	−0.637 *	−0.705 #	2.59
Unknown	MP	5	795	60	60	4	0.900	0.031	24.60	−1.103	−1.045	−1.045	3.134
Asia	CP	55	483	53	60	26	0.875	0.011	5.42	−2.021 *	−1.636	−1.830	−10.006
Australia	CP	44	483	7	7	7	0.732	0.002	1.12	−0.833	−1.496	−1.725	−1.775
Europe	CP	31	483	71	77	19	0.951	0.039	18.76	−0.100	−0.700	−0.712	0.084
Middle East	CP	7	483	57	60	7	1.000	0.053	25.62	0.268	−0.728	−0.764	−0.126
North America	CP	29	483	45	47	9	0.761	0.014	6.91	−1.579	−0.644 **	−0.630 **	2.442
Unknown	CP	5	483	44	45	4	0.900	0.038	18.50	−1.081	−0.298	−0.298	2.636

The table includes the following data: *n*—number of sequences; Sites—number of nucleotides analysed; S—number of segregating/polymorphic sites; Eta—total number of mutations; Var—number of variants; Vd—variant diversity; Pi—nucleotide diversity; k—average number of nucleotide differences; Tajima’s D; FuLiD *—Fu and Li’s D *-test statistic; FuLiF *—Fu and Li’s F *-test statistic; FuFs—Fu’s Fs statistic. Tajima’s D: * *p* > 0.10; ** *p* < 0.01. Fu and Li’s D*-test statistic: #, 0.10 > *p* > 0.05; * *p* < 0.05; ** *p* < 0.02. Fu and Li’s F *-test statistic: #, *p* > 0.10; * *p* < 0.05; ** *p* < 0.02.

## Data Availability

Not applicable.

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
