# Peer review of "Genome Characterisation of the CGMMV Virus Population in Australia—Informing Plant Biosecurity Policy"

_viruses, 2023, doi:10.3390/v15030743_

Round 1
Reviewer 1 Report
The manuscript by Macki et al. entitled: "Genome characterization of the CGMMV virus population in Australia – informing Plant Biosecurity Policy" describes analyses of Australian CGMMV isolates that were compared to isolates from around the world in order to determine possible sources for CGMMV spread in Australia. The authors have conducted sequencing of 35 new CGMMV isolates among the 42 isolates, using primarily high throughput sequencing (HTS) and for several isolates, they have used Oxford Nanopore (ONT) MinION. The samples were collected from various sources: cucurbits, weeds, pollen, seeds, and several locations. Genome sequence alignments, phylogenetic analyses, and Median-joining network analysis were performed. The authors conclude that CGMMV in Australia resulted from a single source.
Comments to authors: This is a comprehensive study of possible sources for CGMMV in Australia.
1. Lines 409 and 423 please explain how it is possible to deduce from phylogenetic analysis of Figure 3 that there was only one Australian clade compared to two clades in Figure 4, when Figure 3 has indeed almost twice the amount of isolates presented in Figure 4 but two major clades are clearly visible.
2. The authors conducted HTS analyses but they did not relate to other viruses that must have been found in the analyses. The co-occurrence of viruses with CGMMV could affect the spread of CGMMV (antagonism or synergism effect) and might change the conclusions regarding the virus's spread.
3. Please correct the reference list. There are several references with the same number.
Reviewer 2 Report
Authors (Mickie et al.) conducted surveys throughout Australia (Northern Territory, Queensland, Western Australia, New South Wales and South Australia), extracted total plant RNA from 42 collected samples, and performed high throughput sequencing. Using the full coding genome sequences obtained from the current study along with all CGMMV full genome sequences available in the GenBank, Authors conducted various phylogenetic and recombination analyses and revealed very high sequence identity among these Australian isolates. A low genetic diversity suggests the CGMMV isolates in Australia were likely resulted from a single source of the virus from multiple introductions.
A similar study on genetic diversity analysis of a large number of CGMMV isolates from California was published recently (Pitman et al., 2022 Plant Disease 106:1713-1722), it would be necessary to refer to that publication in the introduction and discuss on the similarity and different scenario between Australia and the U.S. (CA) CGMMV genetic diversity.
In addition:
L19-20: 42 or 35 new genome sequences of Australian samples? There is a need for clarification, it is not matching between the number of samples in the main text and in the Abstract.
L93-96: Table 1. Simplify the title description by keeping only the first sentence for the table title and the rest statements to be moved to the footnote. Check other tables for the same situation.
L240: Spell out Sequence Demarcation Tool (SDT) at the first mention, please also use its full name in several Table titles.
L340. remove "(in) are placed" in this sentence?
L375-381, Table 4. Keep the first sentence for the title and the rest to be moved to the footnotes.
L385: Discussion, need to discuss your results and those presented in a recent paper in California on CGMMV genetic diversity (Pitman et al., 2022).
